# Instruction-tuned LLMs with World Knowledge are More Aligned to the Human Brain

## Abstract

Instruction-tuning is a widely-adopted method of finetuning that enables large language models (LLMs) to generate output that more closely resembles human responses to natural language queries, in many cases leading to human-level performance on diverse testbeds. However, it remains unclear whether instruction-tuning truly makes LLMs more similar to how humans process language. We investigate the effect of instruction-tuning on LLM-human similarity in two ways: (1) *brain alignment*, the similarity of LLM internal representations to neural activity in the human language system, and (2) *behavioral alignment*, the similarity of LLM and human behavior on a reading task. We assess 25 vanilla and instruction-tuned LLMs across three datasets involving humans reading naturalistic stories and sentences. We discover that instruction-tuning generally enhances brain alignment by an average of $6\%$, but does not have a similar effect on behavioral alignment. To identify the factors underlying LLM-brain alignment, we compute correlations between the brain alignment of LLMs and various model properties, such as model size, various problem-solving abilities, and performance on tasks requiring world knowledge spanning various domains. Notably, we find a strong positive correlation between brain alignment and model size (r = 0.95), as well as performance on tasks requiring world knowledge (r = 0.81). Our results demonstrate that instruction-tuning LLMs improves both world knowledge representations and brain alignment, suggesting that mechanisms that encode world knowledge in LLMs also improve representational alignment to the human brain.

## 1 Introduction

Instruction-tuning is a widely adopted method for finetuning large language models (LLMs) on datasets containing task-specific instructions. This approach enhances their ability to generalize effectively to previously unseen tasks by learning to follow provided instructions (Wang et al., 2022c). Instruction-tuning often costs only a small fraction of compute relative to pretraining (Chung et al., 2022), yet propels pretrained LLMs to incredible performance leaps on reasoning and problem-solving benchmarks. This transformation has enabled LLMs to approach human performance on many tasks, despite using only few (or zero) training examples, as well as to tackle open-world reasoning tasks previously only achievable by humans (Zhang et al., 2023).

In addition to teaching LLMs to understand and follow human instructions, instruction-tuning also improves the ability of LLMs to mimic the ground-truth outputs (often human-written) of the training data. This property allows them to produce more controllable and predictable output that is deemed (1) more desirable by human evaluators on various metrics (Zhang et al., 2023; Chung et al., 2022; Wang et al., 2022b), (2) more aligned to human values (Chia et al., 2023), and (3) more stylistically similar to human outputs (Dasgupta et al., 2022; Safdari et al., 2023).

Consequently, instruction-tuning yields LLMs more similar to humans in both capability and output rememblance. From a neuroscience perspective, this begs the question: **Does instruction-tuning make LLMs more similar to the human language system?** Previous work has shown that models with higher task performance are more aligned to the human language system (Schrimpf et al., 2021; Goldstein et al., 2022; Caucheteux & King, 2022), and, on some datasets, even hit the estimated

noise ceiling.[1] However, there has been no similar study on how instruction-tuning, the method that has enabled powerful LLMs such as ChatGPT, affects alignment to the human language system.

In this work, we explore the impact of instruction-tuning on the alignment between LLMs and the human language system, considering two aspects: (1) *brain alignment*, which assesses how closely LLMs' internal representations match neural activity patterns in the human language system, and (2) *behavioral alignment*, which evaluates the similarity between LLM behavior and human behavior. To conduct this study, both LLMs and human participants are presented with the same language stimuli comprised of naturalistic stories and sentences. For LLMs, we analyze their internal representations and per-word perplexity, while for humans, we use previously collected brain activity data from functional magnetic resonance imaging (fMRI) experiments and per-word reading times.

To measure brain alignment, we use the Brain-Score (Schrimpf et al., 2018) linear predictivity metric, assessing how well LLM representations predict human brain activity in response to the same language stimuli (Jain & Huth, 2018; Toneva & Wehbe, 2019; Schrimpf et al., 2021; Oota et al., 2023), using data from three neural datasets: Pereira et al. (2018), Blank et al. (2014), and Wehbe et al. (2014). To evaluate behavioral alignment, we use a benchmark in Brain-Score which calculates the Pearson correlation between LLM per-word perplexity and human per-word reading times from the Futrell et al. (2018) dataset. Perplexity for LLMs and reading times for humans offer insights into comprehension difficulty (Ehrlich & Rayner, 1981; Hale, 2001; Smith & Levy, 2013), allowing us to examine whether LLMs and humans share similarities in terms of which words and sentences they find challenging or surprising. Because models vary in their brain and behavioral alignment across different architectures and training objectives (Schrimpf et al., 2021), we estimate the effect of instruction-tuning across 17 instruction-tuned LLMs and 8 vanilla LLMs, and report a significant increase in brain alignment by instruction-tuned models compared to vanilla ones.

To investigate *why* instruction-tuning increases alignment to human brain activity, we then estimate the contribution of various LLM properties towards brain alignment. Specifically, we compute Pearson correlations between an LLM's brain alignment and its properties, including next-word prediction (NWP) ability, model size, a range of problem-solving abilities, and world knowledge spanning different domains. The evaluation of the latter two properties is based on the Big-Bench Hard benchmark (BBH) (Suzgun et al., 2022) and the Massive Multi-task Language Understanding benchmark (MMLU) (Hendrycks et al., 2021), respectively.

We report three major findings:

1. Instruction-tuning generally improves the alignment of LLM representations to brain activity, increasing brain alignment by 6.2% on average for the LLMs and neural datasets we tested (Figure 1).

2. Investigating the factors underlying LLM-brain alignment, we find that world knowledge and model size are strongly correlated with brain alignment (r = 0.81 and 0.95 for instruction-tuned models, respectively; Figure 2).

3. Surprisingly, our results generally indicate that instruction-tuning LLMs does not enhance behavioral alignment with human reading times. Furthermore, behavioral alignment on this dataset demonstrates poor correlations with all other measures we investigate, including task performance and model size (Figure 3).

## 2 BACKGROUND & RELATED WORK

**Effect of Instruction-tuning on LLMs.** Instruction-tuning is an effective method for enhancing LLM capability and controllability. It trains LLMs using pairs of human instructions and desired outputs. The benefits of instruction-tuning can be categorized into three key aspects (Zhang et al., 2023): (1) it bridges the disparity between the pretraining objective of LLMs (next-word prediction) and the goal of accurately following human instructions, (2) it achieves greater control and predictability of model behavior compared to standard LLMs, allowing researchers to make them more similar to humans in both capability and output similarity (Chia et al., 2023; Dasgupta et al., 2022; Safdari et al., 2023), and (3) it often costs only a small fraction of compute relative to pre-

---

[1]In fMRI recordings, an upper limit of representational similarity can be computed by sampling from the same participant twice, deducing a threshold defined by the noise level of the data gathering process.

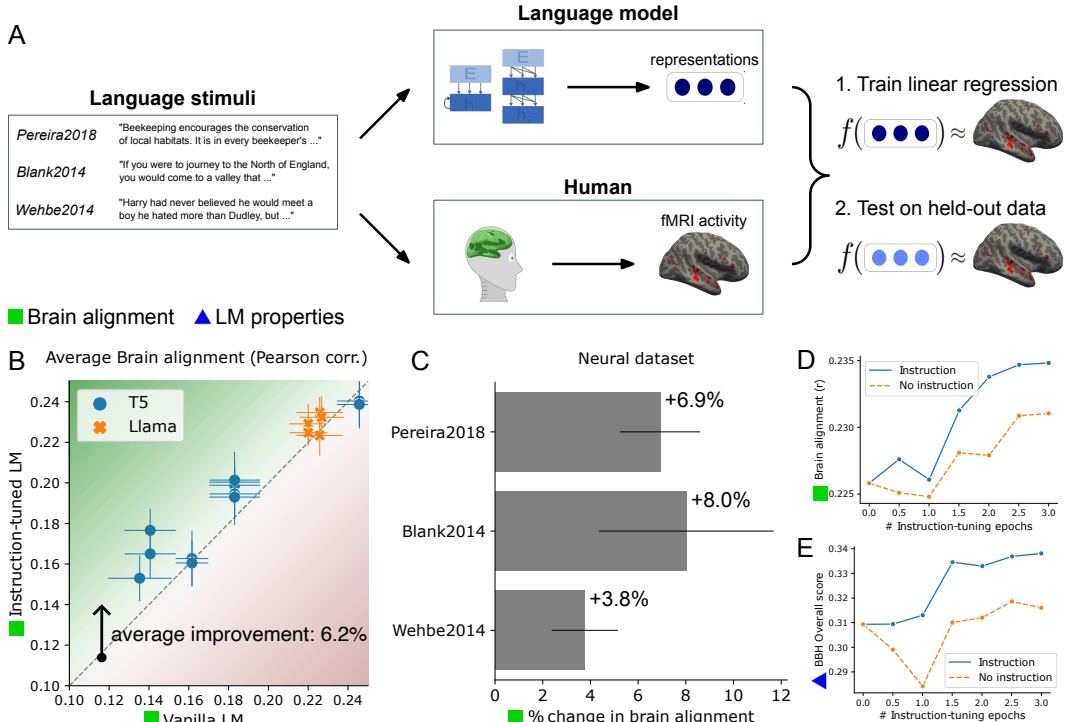

Figure 1: **Instruction-tuning aligns LLM representations to human brain activity. (A)** We evaluate brain alignment as the similarity of an LLM's internal representations to human brain activity, using a linear predictivity metric in Brain-Score. We evaluate 25 vanilla and instruction-tuned LLMs with sizes between 77M and 33B parameters. We use 3 neural datasets of humans reading naturalistic stories and sentences: PEREIRA2018, BLANK2014, and WEHBE2014, and compute the average across the three datasets. **(B)** Instruction-tuning improves average brain alignment by 6.2% on average. Each point above the identity line represents an instruction-tuned LLM that has greater brain alignment than its vanilla counterpart. Error bars (here and elsewhere) represent median absolute deviation over human participants. **(C)** Instruction-tuning generally improves brain alignment on all three neural datasets. **(D)** We instruction-tune LLaMA-7B using the Alpaca dataset ("Instruction" model). We also train an ablation model with the same process and training data, but remove the instruction portion from each training sample ("No Instruction" model). This experiment shows that brain alignment improvements are due to both (1) training data (present in both models) and (2) the process of training LLMs to understand and follow instructions (present only in original model).

training, enabling LLMs to swiftly adapt to target domains (Chung et al., 2022). We contribute to this research area from a neuroscience perspective, by studying whether instruction-tuning makes LLMs more aligned to the human language system in terms of brain and behavioral alignment.

**Effect of Finetuning on Brain alignment.** Prior works have studied how finetuning affects LMs' alignment to human brain activity. These include finetuning on a wide range of downstream NLP tasks (Oota et al., 2022), finetuning to summarize narratives (Aw & Toneva, 2023), and finetuning to directly predict brain activity recordings (Schwartz et al., 2019). These studies aim to use brain alignment to study how finetuning affects LMs and their representations. Our work builds on this line of research by demonstrating that instruction-tuning aligns LLM representations to human brain activity. We also investigate why instruction-tuned LLMs align to brain activity by testing the correlation of brain alignment with various world knowledge domains and problem-solving abilities.

**LM properties linked to Brain alignment.** There is a growing body of work disentangling the contribution of various LM properties towards brain alignment. These include studying how brain alignment is driven by next-word prediction ability (Schrimpf et al., 2021; Caucheteux & King, 2022), multi-word semantics (Merlin & Toneva, 2022), performance on various NLP tasks (Oota et al., 2022), and model size (Antonello et al., 2023). To disentangle the contribution of various

LM properties toward brain alignment, we use instruction-tuned LLMs. They are especially useful as they have been trained to respond to a standard question-answer format, allowing us to evaluate LLMs on a wide array of tasks and in a more fine-grained manner. We expand this area of research by identifying that world knowledge is a key property underlying LLM-brain alignment.

## 3 LANGUAGE MODELS

We evaluate the brain alignment of 25 large language models (LLMs) from two model families: T5 (Raffel et al., 2020) and LLaMa (Touvron et al., 2023). T5 models are encoder-decoder LLMs pretrained on the Colossal Common Crawl Corpus (C4), a corpus of 356 billion tokens, using a masked infilling objective, and then further finetuned on multi-task mixture of unsupervised and supervised tasks converted into a text-to-text format. In our study, we use all five T5 models with sizes between 77M to 11B parameters. LLaMA models (Touvron et al., 2023) are decoder-only LLMs trained on 1.6 trillion tokens from a mixture of corpora including C4, English CommonCrawl, Wikipedia, Github, and more. For LLaMA, we use the 7B, 13B, and 33B parameter versions in our study.

For the instruction-tuned variants of T5 models, we utilize a variety of models finetuned on the FLAN suite (15M examples for 1,836 different tasks accompanied by instructions, Chung et al., 2022), Alpaca (52K instruction-following examples generated through methods inspired by Self-Instruct, Wang et al. (2022a), Taori et al., 2023), and GPT4ALL (437K instruction-following examples generated with GPT-3.5-turbo, Anand et al., 2023) datasets. As for the LLaMa model family, we employ Vicuna's 7B, 13B, and 33B models (Chiang et al., 2023), which are finetuned on user-shared conversations. Additionally, we incorporate the StableVicuna-13B model, which further refines the Vicuna-13B model using reinforcement learning from human feedback (RLHF) (Ouyang et al., 2022) on a range of conversational and instructional datasets. We also use the 7B version of Alpaca (Taori et al., 2023). Additional details about these LLMs can be found in Appendix A.

Table 1: **Example of instruction-tuning training data format: (Instruction, Input, Output)** from the Alpaca dataset (Taori et al., 2023). The input field is optional for certain types of instructions.

| Instruction | Input | Output |
|---|---|---|
| "Write a short paragraph about the given topic." | "The importance of using renewable energy." | "The use of renewable energy is growing rapidly in relevance and ..." |

## 4 BRAIN ALIGNMENT

Brain alignment refers to the method of evaluating the representational similarity between LLMs and human brain activity (Figure 1). This relies on fMRI recordings of human subjects while they read specific language stimuli on potentially any topic (here: Pereira et al., 2018; Blank et al., 2014; Wehbe et al., 2014). In brain alignment studies, the same language stimuli from prior brain recordings are provided as input to LLMs, whose intermediate layer activations are recorded to extract model representations of the language stimuli. To study the alignment of LLM and human data, we follow a general approach previously used in several works (Schrimpf et al., 2018; 2021; Jain & Huth, 2018; Toneva & Wehbe, 2019; Oota et al., 2023; Aw & Toneva, 2023). Specifically, we use the linear predictivity metric implemented in Brain-Score (Schrimpf et al., 2020, Figure 1), first training a linear function to predict fMRI voxels associated with the human language system using LLM representations as input features. We then apply this linear function to held-out brain activity data from the original corpus of recordings, and evaluate the brain alignment of the LLM as the Pearson correlation between the predicted and actual brain activity data. For each LLM, we evaluate its brain alignment for every layer (e.g., LLaMA-7B has 32 layers), and use the highest value as the LLM's brain alignment value, following Schrimpf et al. (2018).

**Datasets** We use three fMRI datasets to measure the brain alignment of LLMs. Each dataset involves a different set of human participants and uses a different set of language stimuli.

PEREIRA2018 (experiments 2 and 3 from Pereira et al., 2018): In experiment 2, nine participants read 384 sentences organized into 96 text passages. In experiment 3, six participants read 243 sentences in 72 text passages. Each sentence was displayed for four seconds on a screen.

BLANK2014 (Blank et al., 2014): The data consists of fMRI recordings of 5 human participants listening to naturalistic stories from the Natural Stories Corpus (Futrell et al., 2018).

WEHBE2014 (Wehbe et al., 2014): The data includes fMRI recordings of 8 human participants reading chapter 9 of the book *Harry Potter and the Sorceror's Stone* (Rowling et al., 1998). Participants read the chapter at a fixed interval of one word every 0.5 seconds.

## 4.1 INSTRUCTION-TUNING ALIGNS LLM REPRESENTATIONS TO HUMAN BRAIN ACTIVITY

First, we study the effect of instruction-tuning on LLM brain alignment. We compute each LLM's average brain alignment as the mean of its brain alignment on the 3 neural datasets. We find that instruction-tuning improves alignment by an average of 6.2% across all tested LLMs (Figure 1B). This holds across all three neural datasets, with average improvements of +6.9% on PEREIRA2018, +8.0% improvement on BLANK2014, and +3.8% on WEHBE2014 (Figure 1C). This provides a strong signal that instruction-tuning improves brain alignment. Moreover, a smaller instruction-tuned model can attain higher brain alignment than a larger model from the same family that has not been instruction-tuned (e.g., Alpaca-7B v.s. LLaMa-13B, see detailed results in Appendix E).

To longitudinally study how instruction-tuning aligns LLM representations to brain activity, we separately instruction-tune a LLaMA-7B model on the Stanford Alpaca instruction dataset (Taori et al., 2023) for 3 epochs. By evaluating checkpoints regularly during training, we find that instruction-tuning progressively improves brain alignment (Figure 1D). We also perform an ablation study to disambiguate the effect on brain alignment of (1) the instruction-following ability provided by the instruction-tuning step and (2) added training data. We fine-tune LLaMA-7B with the same process and data, but remove the instruction portion from each training sample. We observe that brain alignment of this ablated model increases during fine-tuning but stays lower than its instruction-following counterpart (Figure 1D). This shows that brain alignment improvements are due to both factors.

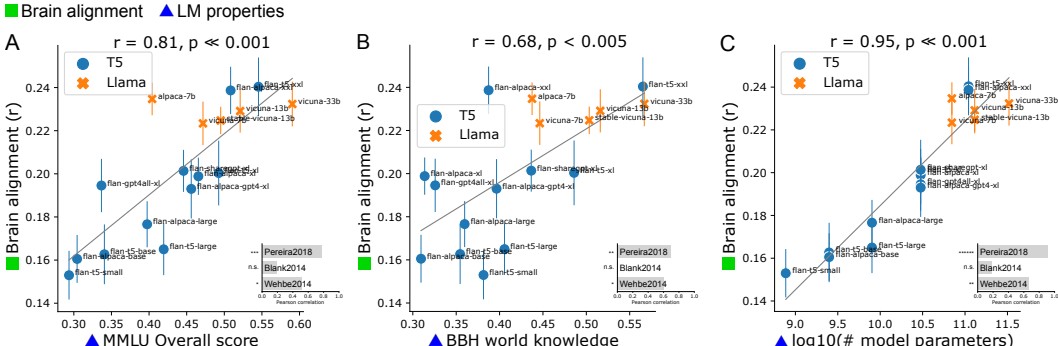

Figure 2: **World knowledge and model size are important factors underlying LLM-brain alignment.** To identify factors underlying brain alignment, we test Pearson correlations between brain alignment and various LLM properties, such as model size, world knowledge in various domains (MMLU benchmark), and various types of problem-solving abilities (BBH benchmark). Insets display results on individual datasets, with stars reflecting statistical significance (n.s. = p > 0.05, * = p < 0.05, ** = p < 0.005, etc.) **(A)** Brain alignment is significantly and strongly correlated with world knowledge as evaluated by the MMLU Overall score (r = 0.81), which reports the mean performance across all MMLU subjects. **(B)** Brain alignment is significantly and strongly correlated with performance on the world knowledge task category in BBH (r = 0.68). **(C)** Brain alignment is significantly and strongly correlated with model size (logarithm of number of parameters) (r = 0.95).

## 4.2 FACTORS UNDERLYING LLM-BRAIN ALIGNMENT

To identify factors underlying the representational similarity between LLMs and human brains, we compute the Pearson correlation between LLM brain alignment and various properties of LLMs: performance on a benchmark involving different reasoning abilities (BBH benchmark; Suzgun et al., 2022), performance on a benchmark requiring domain-specific world knowledge (MMLU; Hendrycks et al., 2021), language modeling ability, and model size.

**World Knowledge and Reasoning abilities**  MMLU is designed to measure the world knowledge of LLMs across many subject domains. It contains 57 tasks, categorized by the subject domain of world knowledge tested: STEM, Humanities, Social Sciences, and Others (a broad category ranging from finance to marketing to professional medicine). BBH contains 23 tasks, grouped into four categories based on the type of problem-solving ability tested: Algorithmic and Multi-Step Arithmetic Reasoning; Natural Language Understanding; Use of World Knowledge; and Multilingual Knowledge and Reasoning. For both benchmarks, we use category classifications from the original papers. We measure the performance of LLMs on BBH and MMLU using the `instruct-eval` repository[2] with default settings (3-shots, 5-shots respectively) and preset prompts. We measure the Pearson correlation (and its p-value) between LLM brain alignment and performance on each category of the MMLU and BBH benchmarks. We obtain p-values after performing false discovery rate correction, ensuring our results are significant and not due to random chance.

We find that brain alignment is significantly and strongly correlated with world knowledge. On the MMLU benchmark, we observe a high correlation between brain alignment scores and the MMLU Overall score (r = 0.81), which reports the mean performance across all world knowledge subject domains on MMLU (Figure 2A). Similarly, brain alignment is also significantly and strongly correlated with the mean performance on tasks included in the world knowledge category of the BBH benchmark (r = 0.68; Figure 2B). Interestingly, we do not find strong correlations with other dimensions of the BBH benchmark (e.g., Algorithmic reasoning and Multilingual reasoning, see Table 2), though this could also be due to limitations of the tested models (most are primarily pretrained on English language data), as indicated by their low raw performance scores on some tasks. Overall, our results provide a strong signal that more accessible representations of world knowledge are a key factor in aligning LLM representations to human brain activity.

**Language Modeling Ability**  Prior works have shown correlations between brain alignment and next-word prediction (NWP) ability (Caucheteux & King, 2022; Schrimpf et al., 2021). We find similar results for correlation between brain alignment and NWP loss (r = -0.54, Appendix G). Interestingly, the strength of correlation between brain alignment and world knowledge performance (r = 0.81) is greater than that between brain alignment and NWP loss (r = -0.54). This result suggests that world knowledge understanding is a better predictor of brain alignment than NWP ability.

**Model Size**  Finally, we find that brain alignment is significantly and strongly correlated with model size (r = 0.95), as measured by the logarithm of the number of model parameters (Figure 2C). Schrimpf et al. (2021) observe such a pattern for language models, and we find the pattern holds for instruction-tuned models, and models trained at a larger scale than their study (7B+ parameters).

However, model size alone does not determine brain alignment. Our results show that smaller instruction-tuned LLMs can have greater brain alignment than larger vanilla models. For example, LLaMA-13B obtains brain alignment of 0.220, Vicuna-13B obtains 0.229, LLaMA-33B obtains 0.227, and Vicuna-33B obtains 0.232. Hence, Vicuna-13B has greater brain alignment than LLaMA-33B despite being less than 40% its size. We observe a similar trend in another four models: T5-base, Flan-T5-base, T5-large, Flan-T5-large. Also, prior works have shown that large random models achieve poor brain alignment (Schrimpf et al., 2021). These results demonstrate that there are LLM properties aside from model size that contribute significantly to brain alignment.

**Datasets**  We observe significant correlations between brain alignment and world knowledge and model size on PEREIRA2018 and WEHBE2014, but surprisingly not BLANK2014. We hypothesize that it may be because BLANK2014 has a small participant size (N = 5) that leads to greater noise in the results, compared to PEREIRA2018 and WEHBE2014.

## 5 BEHAVIORAL ALIGNMENT

In the previous section, we show that instruction-tuning aligns the internal representations of LLMs to human *brain recordings* (Section 4.1). In this section, we explore whether instruction-tuning also aligns LLM behavior to human *behavior*.

Following the approach previously proposed by Schrimpf et al. (2021) and implemented in the Brain-Score package (Schrimpf et al., 2020), we measure behavioral alignment by evaluating the

---

[2]https://github.com/declare-lab/instruct-eval

Table 2: **Brain alignment strongly correlates with world knowledge across all subject domains in MMLU, as well as the world knowledge category in BBH.** At the same time, brain alignment is not significantly correlated with all other types of problem-solving abilities in BBH (e.g., algorithmic or multilingual reasoning). We obtain p-values after performing false discovery rate correction, ensuring our results are significant and not due to random chance.

| Task category | Brain Alignment Correlation ($r$) | corrected $p$-value | Number of tasks | Average Model Performance |
|---|---|---|---|---|
| MMLU – Overall Score | **0.809** | **0.000329** | 57 | 0.36 |
| MMLU – STEM | **0.792** | **0.000343** | 18 | 0.28 |
| MMLU – Humanities | **0.791** | **0.000343** | 13 | 0.34 |
| MMLU – Social Sciences | **0.807** | **0.000329** | 12 | 0.41 |
| MMLU – Others | **0.809** | **0.000329** | 14 | 0.40 |
| BBH – Overall score | 0.384 | 0.177 | 23 | 0.28 |
| BBH – Algorithmic reasoning | 0.194 | 0.558 | 8 | 0.22 |
| BBH – Language understanding | 0.163 | 0.585 | 3 | 0.43 |
| BBH – World knowledge | **0.679** | **0.005** | 5 | 0.36 |
| BBH – Multilingual reasoning | -0.035 | 0.895 | 1 | 0.19 |
| BBH – Others | 0.478 | 0.083 | 6 | 0.27 |

similarity between LLM per-word perplexity and human per-word reading times, given the same language stimuli (Figure 3A). We use the self-paced reading times dataset from Futrell et al. (2018), consisting of the reading times of 179 human participants recorded while they were visually presented with 10 naturalistic stories. We provide language stimuli from this data as input to LLMs and measure their per-word perplexity. Finally, we evaluate behavioral alignment by computing the Pearson correlation between per-word LLM perplexity and per-word human reading times.

Using the same set of models as in the previous section, we compare the behavioral alignment of each instruction-tuned LLM against its vanilla counterpart. Our results generally indicate that instruction-tuning LLMs does not improve behavioral alignment to human reading times (Figure 3B). For half of the LLMs tested, it results in no change or reduced behavioral alignment. Then, we test the correlation between LLM behavioral alignment and model size, next-word prediction ability, various reasoning abilities (as measured by performance on the BBH benchmark), and world knowledge across various domains (as measured by performance on the MMLU benchmark). Contrary to our findings on the correlation between brain alignment and model size and world knowledge (Section 4.2), we do not find that these factors are correlated with the behavioral alignment of models: world knowledge in Figure 3C (r = 0.08, p = 0.76), model size in Figure 3D (r = 0.26, p = 0.31), next-word prediction loss for T5 models in Figure 3E (r = -0.2, p = 0.54), and next-word prediction loss for LLaMA models in Figure 3F (r = 0.68, p = 0.21). We discuss our interpretations of these results and possible explanations in Section 6.2.

# 6 DISCUSSION

## 6.1 IMPLICATIONS FOR NLP: BUILDING LLMS

**Using brain alignment to interpret LLMs.** Representations in the human brain support a wide range of downstream tasks that current models still struggle on. Prior works have used brain activity results to interpret neural networks (Dong & Toneva, 2023) and build more performant models (Dapello et al., 2020; 2022; Safarani et al., 2021). Instruction-tuning has emerged as a breakthrough technique to improve LLM abilities, quality of outputs, and allow LLMs to adapt to new tasks with minimal task-specific training. However, the manner in which instruction-tuning alters the internal representations of LLMs to achieve these improvements remains an open question. Brain activity data provides a neuroscientific angle to investigate this question. Our results show that instruction-tuning LLMs improves both world knowledge representations and human brain alignment. They suggest that instruction-tuning improves the ability of an LLM to retrieve world knowledge stored in its model weights and that the mechanisms that encode world knowledge in LLMs also improve

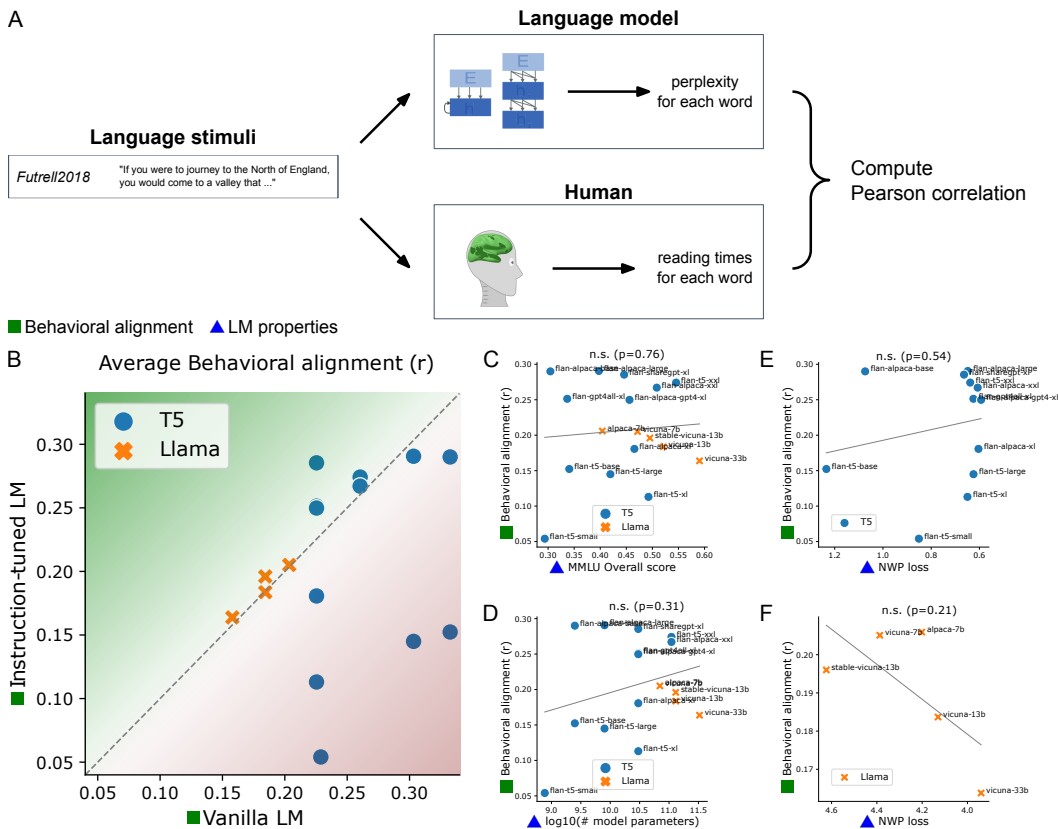

Figure 3: **Instruction-tuning LLMs generally does not improve behavioral alignment to human reading times. Furthermore, behavioral alignment correlates poorly with all other tested measures:** world knowledge, model size, and next-word prediction (NWP) ability. **(A)** We present the same language stimuli to LLMs and human participants, using the *Futrell2018* benchmark in Brain-Score, which contains naturalistic stories. We compute the behavioral alignment as the Pearson correlation between LLM perplexity for each word and human reading times for each word. **(B)** Instruction-tuning does not generally improve behavioral alignment. Furthermore, behavioral alignment is poorly and not significantly correlated with all other measures: **(C)** world knowledge on MMLU (r = 0.08, p = 0.76), **(D)** model size (r = 0.26, p = 0.31), **(E)** NWP loss for T5 models (r = -0.2, p = 0.54), and **(F)** NWP loss for LLaMA models (r = 0.68, p = 0.21).

representational alignment to the human brain. Overall, our results highlight the importance of integrating world knowledge in the development of future LLMs.

## 6.2 IMPLICATIONS FOR NEUROSCIENCE: STUDYING LLM-HUMAN ALIGNMENT

**Instruction-tuned LLMs are useful for studying LLM properties underlying brain and behavioral alignment.** To identify why LLM and human brains exhibit representational similarities, prior work has mostly focused on high-level properties such as model size (Antonello et al., 2023), and external behaviors such as predicting missing words Schrimpf et al. (2021); Caucheteux & King (2022). However, a key to understanding these similarities is to identify lower-level or internal properties of LLMs that underlie brain alignment. This includes the amount of knowledge LLMs contain, e.g., factual (AlKhamissi et al., 2022) and commonsense (Sap et al., 2020; Bosselut et al., 2019). Our work shows that we can harness instruction-tuned LLMs for this purpose as they have been trained to respond to a general question format, allowing us to evaluate LLMs in a more fine-grained manner. This allows the study of both internal (e.g., knowledge) and external (e.g., behavior) properties of LLMs, and how they correlate with brain and behavioral alignment.

**Examining more dimensions of behavior.** To evaluate behavioral alignment, our work and many prior works compare LM and human next-word surprisal on reading tasks (Wilcox et al., 2020; Schrimpf et al., 2021; Eghbal A. Hosseini et al., 2023). This evaluates only a single dimension of LM and human behavior (per-word perplexity and reading times). On the models we test here, behavioral alignment is not significantly correlated with model size, world knowledge, or next-word prediction ability. While next-word prediction performance correlates with alignment to human reading times across a broad band of models (Schrimpf et al., 2021), this trend does not hold up in recent Transformer-based LMs (Oh & Schuler, 2023), having a surprising negative correlation with parameter count (Oh et al., 2022). Our results highlight the need to create more benchmarks to expand the dimensions of behavior examined for both LLMs and humans, in order to holistically evaluate LLM behavior, as well as LLM-human behavioral alignment.

**Brain alignment datasets with humans performing diverse tasks.** Our work studies brain alignment using neural datasets that are limited to humans reading naturalistic stories and sentences in English. Unfortunately, there does not exist brain activity data for human participants attempting the BBH and MMLU benchmarks. This may explain why brain alignment is not significantly correlated with many categories of problem-solving on BBH, e.g., language understanding. In the future, we hope to study brain alignment with human participants performing more diverse sets of tasks, e.g., reading computer program code (Ivanova et al., 2020). This can identify more factors underlying LLM-brain alignment, and provide insights into how brain activity and the human language system may be shaped by various forms of problem-solving. Furthermore, for the neural datasets in our work, many of the larger models exceed the noise ceiling estimates (Appendix E), highlighting the need for more neural datasets (with better ways of computing noise ceiling estimates).

**World knowledge shapes brain activity.** Our results show that world knowledge is a key factor in aligning LLM representations to human brain activity. LLMs with greater world knowledge across all tested subject domains produce representations that align more closely to human brain activity. Analogously, this suggests that world knowledge influences human brain activity, shaping the language comprehension systems in the brain.

### 6.3 LIMITATIONS AND FUTURE WORK

**Limited model families.** A model's performance on MMLU and BBH is affected not only by the quantity of knowledge it possesses, but also by its ability to follow each benchmark's instruction format. Hence, we restricted our study to two model families (T5 and LLaMA). This attempts to ensure that the models are trained on a similar instruction format, so they have similar ability to follow the instruction format of MMLU and BBH. Thus, any differences in their MMLU or BBH performance would point to differences in the quantity of knowledge they possess. However, we hope to add a greater variety of LLMs in future work to strengthen the generalizability of our claims.

**Computational cost.** We investigate 25 LLMs, with the largest models having 33B parameters, on many datasets: brain alignment (PEREIRA2018, BLANK2014, WEHBE2014), behavioral alignment (FUTRELL2018), next-word prediction (WikiText-2) and other evaluations (BBH, MMLU). Unfortunately, this makes it computationally expensive to replicate our experiments. In the hopes of addressing this concern, we provide our full results in Appendices E, F, G, H, and I.

## 7 CONCLUSIONS

We investigate whether instruction-tuning improves the alignment of LLMs to the human language system. We evaluate 25 LLMs with parameter sizes ranging from 77 million to 33 billion, across three neural datasets of humans reading naturalistic stories and sentences. We find that instruction-tuning generally improves the alignment of LLM representations to brain activity. Delving into the factors underlying LLM-brain alignment, we discover that world knowledge and model size are key determinants of brain alignment. This suggests that world knowledge helps shape representations in the human language system, and highlights the significance of integrating world knowledge in the development of future LLMs.

REPRODUCIBILITY STATEMENT

All models and code repositories utilized in this study are open-source and their corresponding links are provided in Appendix B and D for full transparency and reproducibility. For the calculation of brain and behavioral alignment, we relied on the Brain-Score repository (`www.github.com/brain-score/language`), which serves as a publicly accessible resource for conducting these assessments. We encourage researchers interested in replicating our findings to refer to the provided links and consult the Brain-Score repository for further details on datasets and the evaluation process. For measuring the performance of LLMs on BBH and MMLU benchmarks, we use the broadly-used instruct-eval repository (`https://github.com/declare-lab/instruct-eval`) with default settings.

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

# A  LANGUAGE MODELS: PARAMETER COUNT AND NUMBER OF LAYERS

Table 3: **Parameter count and number of layers for all 25 vanilla and instruction-tuned LLMs.** The upper part contains encoder-decoder models of the T5 family, the lower parts decoder-only models of the LLaMA family. For the parameter count, "M" refers to million and "B" refers to billion. The number of layers for T5 models is a sum of the number of encoder and decoder layers.

| Model | Parameter Count | Number of Layers |
|---|---|---|
| t5-small | 77 M | 16 |
| flan-t5-small | 77 M | 16 |
| t5-base | 250 M | 24 |
| flan-t5-base | 250 M | 24 |
| flan-alpaca-base | 250 M | 24 |
| t5-large | 800 M | 48 |
| flan-t5-large | 800 M | 48 |
| flan-alpaca-large | 800 M | 48 |
| t5-xl | 3 B | 48 |
| flan-t5-xl | 3 B | 48 |
| flan-alpaca-xl | 3 B | 48 |
| flan-gpt4all-xl | 3 B | 48 |
| flan-sharegpt-xl | 3 B | 48 |
| flan-alpaca-gpt4-xl | 3 B | 48 |
| t5-xxl | 11 B | 48 |
| flan-t5-xxl | 11 B | 48 |
| flan-alpaca-xxl | 11 B | 48 |
| llama-7b | 7 B | 32 |
| alpaca-7b | 7 B | 32 |
| vicuna-7b | 7 B | 32 |
| llama-13b | 13 B | 40 |
| vicuna-13b | 13 B | 40 |
| stable-vicuna-13b | 13 B | 40 |
| llama-33b | 33 B | 60 |
| vicuna-33b | 33 B | 60 |

# B LANGUAGE MODELS: LINKS TO MODELS WEIGHTS

Table 4: **Link to model weights for all 25 vanilla and instruction-tuned LLMs.** The upper part contains encoder-decoder models of the T5 family, the lower parts decoder-only models of the LLaMA family. We provide these links for reproducibility purposes.

| Model | Link to model weights |
|---|---|
| t5-small | www.huggingface.co/google/t5-v1_1-small |
| flan-t5-small | www.huggingface.co/google/flan-t5-small |
| t5-base | www.huggingface.co/google/t5-v1_1-base |
| flan-t5-base | www.huggingface.co/google/flan-t5-base |
| flan-alpaca-base | www.huggingface.co/declare-lab/flan-alpaca-base |
| t5-large | www.huggingface.co/google/t5-v1_1-large |
| flan-t5-large | www.huggingface.co/google/flan-t5-large |
| flan-alpaca-large | www.huggingface.co/declare-lab/flan-alpaca-large |
| t5-xl | www.huggingface.co/google/t5-v1_1-xl |
| flan-t5-xl | www.huggingface.co/google/flan-t5-xl |
| flan-alpaca-xl | www.huggingface.co/declare-lab/flan-alpaca-xl |
| flan-gpt4all-xl | www.huggingface.co/declare-lab/flan-gpt4all-xl |
| flan-sharegpt-xl | www.huggingface.co/declare-lab/flan-sharegpt-xl |
| flan-alpaca-gpt4-xl | www.huggingface.co/declare-lab/flan-alpaca-gpt4-xl |
| t5-xxl | www.huggingface.co/google/t5-v1_1-xxl |
| flan-t5-xxl | www.huggingface.co/google/flan-t5-xxl |
| flan-alpaca-xxl | www.huggingface.co/declare-lab/flan-alpaca-xxl |
| llama-7b | www.github.com/facebookresearch/llama |
| alpaca-7b | www.github.com/tatsu-lab/stanford_alpaca |
| vicuna-7b | www.huggingface.co/lmsys/vicuna-7b-v1.3 |
| llama-13b | www.github.com/facebookresearch/llama |
| vicuna-13b | www.huggingface.co/lmsys/vicuna-13b-v1.3 |
| stable-vicuna-13b | www.huggingface.co/CarperAI/stable-vicuna-13b-delta |
| llama-33b | www.github.com/facebookresearch/llama |
| vicuna-33b | www.huggingface.co/lmsys/vicuna-33b-v1.3 |

## C  MMLU AND BBH BENCHMARKS

MMLU is designed to measure knowledge from many domains Hendrycks et al. (2021). It contains 57 tasks, categorized by the subject domain of world knowledge tested: STEM, Humanities, Social Sciences, and Others. The STEM category includes questions on computer science, physics, mathematics, etc. The Humanities category includes questions on philosophy, law, history, etc. The Social Sciences category includes questions on politics, sociology, economics, geography, etc. The Others category includes questions on business topics such as finance, accounting, as well as general knowledge of global facts.

BBH is designed to evaluate various problem-solving and reasoning abilities of LLMs (Suzgun et al., 2022). BBH contains 23 tasks, categorized by the type of problem-solving ability tested: (1) Algorithmic and Multi-Step Arithmetic Reasoning, (2) Natural Language Understanding, (3) Use of World Knowledge, and (4) Multilingual Knowledge and Reasoning. The world knowledge category of BBH contains tasks that test for factual and general knowledge. Tasks requiring factual knowledge include: "Sports Understanding" and "Movie Recommendation". Tasks requiring general knowledge include: "Causal Judgement", which tests knowledge about causal-reasoning suppositions, and "Ruin Names", which requires knowledge about human perception and usage of humor in the English language.

For both benchmarks, we adopt the same category classification as used in their original papers. We measure the performance of LLMs on BBH and MMLU using the `instruct-eval` repository[3] with default settings (3-shots, 5-shots respectively) and preset prompts.

## D  CODE REPOSITORIES

We use the Brain-Score repository to evaluate brain alignment for the PEREIRA2018 and BLANK2014 datasets, as well as behavioral alignment for the FUTRELL2018 dataset. Link: `www.github.com/brain-score/language`.

We use an open-source repository to evaluate brain alignment for the WEHBE2014 dataset. Link: `www.github.com/awwkl/brain_language_summarization`, which builds on `www.github.com/mtoneva/brain_language_nlp`.

We use Instruct-Eval repository to evaluate MMLU and BBH scores. Link: `www.github.com/declare-lab/instruct-eval`.

We use Stanford Alpaca repository for instruction-tuning. Link: `www.github.com/tatsu-lab/stanford_alpaca`).

---

[3]`https://github.com/declare-lab/instruct-eval`

# E    RESULTS FOR BRAIN ALIGNMENT

Table 5: **Brain alignment results for all 25 vanilla and instruction-tuned LLMs.** We provide these results for reproducibility purposes.

|  | PEREIRA2018 | BLANK2014 | WEHBE2014 | Average |
|---|---|---|---|---|
| t5-small | 0.166 | 0.168 | 0.071 | 0.135 |
| flan-t5-small | 0.202 | 0.178 | 0.079 | 0.153 |
| t5-base | 0.222 | 0.188 | 0.074 | 0.162 |
| flan-t5-base | 0.234 | 0.178 | 0.076 | 0.163 |
| flan-alpaca-base | 0.227 | 0.179 | 0.076 | 0.161 |
| t5-large | 0.270 | 0.082 | 0.071 | 0.141 |
| flan-t5-large | 0.311 | 0.104 | 0.080 | 0.165 |
| flan-alpaca-large | 0.322 | 0.126 | 0.082 | 0.177 |
| t5-xl | 0.285 | 0.192 | 0.072 | 0.183 |
| flan-t5-xl | 0.314 | 0.215 | 0.072 | 0.200 |
| flan-alpaca-xl | 0.312 | 0.209 | 0.075 | 0.199 |
| flan-gpt4all-xl | 0.300 | 0.206 | 0.078 | 0.195 |
| flan-sharegpt-xl | 0.323 | 0.211 | 0.070 | 0.201 |
| flan-alpaca-gpt4-xl | 0.302 | 0.205 | 0.073 | 0.193 |
| t5-xxl | 0.343 | 0.297 | 0.096 | 0.246 |
| flan-t5-xxl | 0.350 | 0.268 | 0.103 | 0.240 |
| flan-alpaca-xxl | 0.346 | 0.268 | 0.102 | 0.239 |
| llama-7b | 0.405 | 0.154 | 0.118 | 0.226 |
| alpaca-7b | 0.420 | 0.167 | 0.118 | 0.235 |
| vicuna-7b | 0.399 | 0.152 | 0.119 | 0.223 |
| llama-13b | 0.412 | 0.133 | 0.115 | 0.220 |
| vicuna-13b | 0.423 | 0.148 | 0.116 | 0.229 |
| stable-vicuna-13b | 0.415 | 0.144 | 0.115 | 0.225 |
| llama-33b | 0.426 | 0.145 | 0.109 | 0.227 |
| vicuna-33b | 0.436 | 0.156 | 0.105 | 0.232 |

Table 6: **Noise ceiling estimates for all 3 neural datasets.** For PEREIRA2018 and BLANK2014, noise ceiling estimates are computed using the Brain-Score repository, with details provided in Schrimpf et al. (2021). For WEHBE2014, noise ceiling estimates are also computed using a similar procedure.

|  | PEREIRA2018 | BLANK2014 | WEHBE2014 | Average |
|---|---|---|---|---|
| Noise ceiling | 0.359 | 0.210 | 0.104 | 0.224 |

## F RESULTS FOR NEXT-WORD PREDICTION, MMLU, BBH

Table 7: **WikiText-2 NWP loss, MMLU Overall Score, and BBH Overall Score for all instruction-tuned LLMs.** Results for vanilla LLMs are not shown as they are not adapted for the question formats in the MMLU and BBH benchmarks. We provide these results for reproducibility purposes.

|  | WikiText-2 NWP Loss | MMLU Overall Score | BBH Overall Score |
|---|---|---|---|
| flan-t5-small | 0.851 | 0.294 | 0.287 |
| flan-t5-base | 1.235 | 0.341 | 0.308 |
| flan-alpaca-base | 1.074 | 0.304 | 0.266 |
| flan-t5-large | 0.625 | 0.419 | 0.370 |
| flan-alpaca-large | 0.648 | 0.397 | 0.276 |
| flan-t5-xl | 0.650 | 0.493 | 0.402 |
| flan-alpaca-xl | 0.604 | 0.466 | 0.270 |
| flan-gpt4all-xl | 0.625 | 0.337 | 0.212 |
| flan-sharegpt-xl | 0.664 | 0.446 | 0.363 |
| flan-alpaca-gpt4-xl | 0.593 | 0.456 | 0.348 |
| flan-t5-xxl | 0.638 | 0.545 | 0.443 |
| flan-alpaca-xxl | 0.607 | 0.508 | 0.229 |
| alpaca-7b | 4.201 | 0.404 | 0.328 |
| vicuna-7b | 4.387 | 0.472 | 0.331 |
| vicuna-13b | 4.130 | 0.521 | 0.387 |
| stable-vicuna-13b | 4.623 | 0.495 | 0.380 |
| vicuna-33b | 3.940 | 0.590 | 0.426 |

**Notes on comparing next-word prediction (NWP) loss across model families.** The T5 and LLaMA models belong to separate model families. We wish to caution that comparing next-word prediction loss across different model families may not be meaningful. This is due to several reasons related to architectural differences, training methodologies, and objectives. (1) Architecture: T5 models have an encoder-decoder architecture while LLaMA models have a decoder-only architecture. (2) Training Objectives: The T5 models were trained on supervised and unsupervised tasks, while the LLaMA models were trained only on unsupervised text 3. (3) Loss computation: The loss functions for both model families are computed differently, making it inappropriate to directly compare their loss values. (4) Evaluation Metrics: Next-word prediction loss is just one metric, and it may not capture the overall language understanding capabilities of a model. Hence, we additionally evaluate these LLMs' alignment to human brain activity, as well as their performance on problem-solving abilities (BBH) and tasks requiring world knowledge (MMLU). In summary, while NWP loss is a valuable metric for evaluating language models within the same family or architecture, comparing across different model families may not be meaningful.

# G   RESULTS FOR CORRELATIONS OF BRAIN ALIGNMENT WITH LLM PROPERTIES

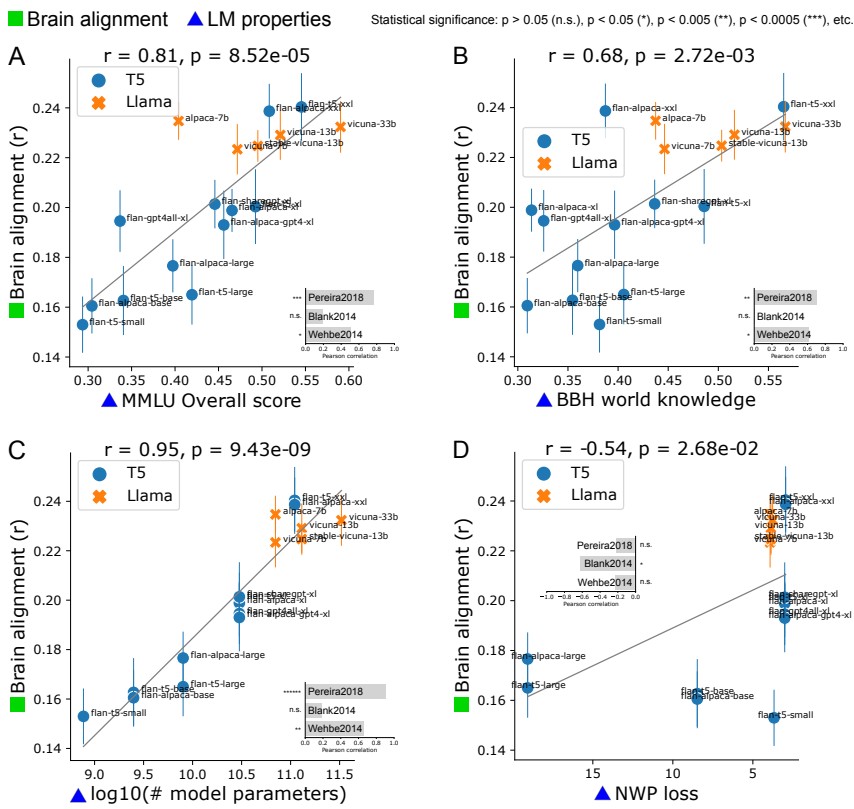

Figure 4: **Correlation between brain alignment and various LLM properties:** (A) MMLU benchmark global score, (B) BBH benchmark score with only world knowledge tasks, (C) number of parameters of the model, and (D) Next word prediction (NWP) performance.

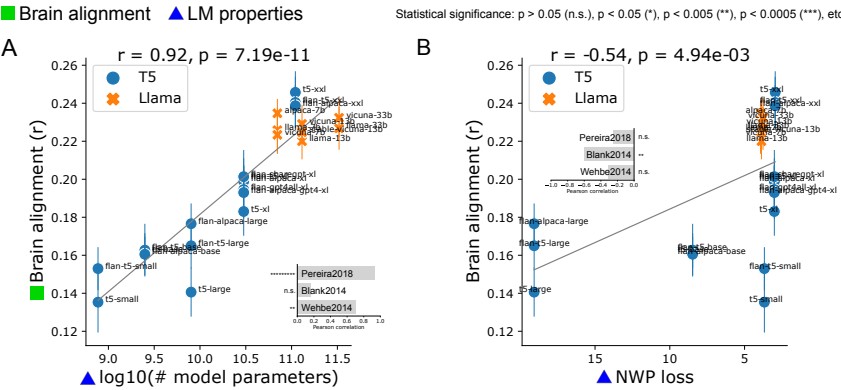

Figure 5: **Correlation between brain alignment and various LLM properties for all 25 LLMs:** (A) number of parameters of the model, and (B) Next word prediction (NWP) performance.

## H   RESULTS FOR BEHAVIORAL ALIGNMENT

Table 8: **Behavioral alignment results for all 25 vanilla and instruction-tuned LLMs.** We provide these results for reproducibility purposes.

|                    | FUTRELL2018 |
|--------------------|-------------|
| t5-small           | 0.229       |
| flan-t5-small      | 0.054       |
| t5-base            | 0.333       |
| flan-t5-base       | 0.152       |
| flan-alpaca-base   | 0.290       |
| t5-large           | 0.303       |
| flan-t5-large      | 0.145       |
| flan-alpaca-large  | 0.291       |
| t5-xl              | 0.225       |
| flan-t5-xl         | 0.113       |
| flan-alpaca-xl     | 0.181       |
| flan-gpt4all-xl    | 0.251       |
| flan-sharegpt-xl   | 0.285       |
| flan-alpaca-gpt4-xl| 0.250       |
| t5-xxl             | 0.260       |
| flan-t5-xxl        | 0.274       |
| flan-alpaca-xxl    | 0.267       |
| llama-7b           | 0.204       |
| alpaca-7b          | 0.206       |
| vicuna-7b          | 0.205       |
| llama-13b          | 0.184       |
| vicuna-13b         | 0.184       |
| stable-vicuna-13b  | 0.196       |
| llama-33b          | 0.158       |
| vicuna-33b         | 0.164       |

Table 9: **Noise ceiling estimates for the FUTRELL2018 reading-times dataset.** Noise ceiling estimates are computed using the Brain-Score repository, with details provided in Schrimpf et al. (2021).

|               | FUTRELL2018 |
|---------------|-------------|
| Noise ceiling | 0.858       |

# I RESULTS FOR INSTRUCTION-TUNING LLaMA-7B ON ALPACA DATASET

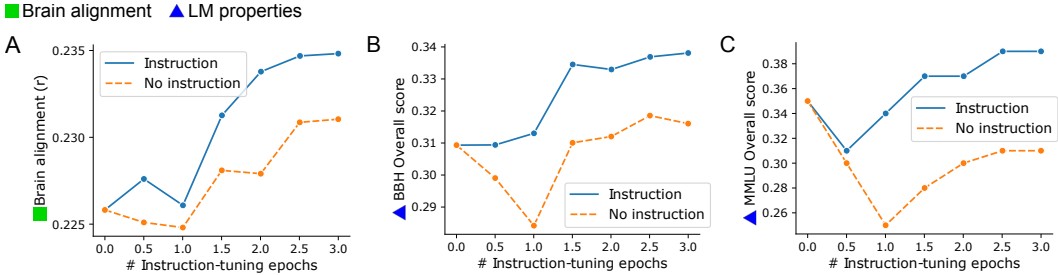

Figure 6: **Improvements in brain alignment from instruction-tuning are due to both additional training data, as well as training to understand and follow instructions.**

**Instruction model**   We instruction-tune LLaMA-7B on the Stanford Alpaca dataset (Taori et al., 2023) using the default training process, following the code in `www.github.com/tatsu-lab/stanford_alpaca`. In particular, the model is instruction-tuned using 52K instruction-following examples generated through methods inspired by Self-Instruct (Wang et al., 2022a)). This model is labeled "Instruction" in Figure 6.

**No instruction model (Ablation)**   We also train an ablation model with the same process and training data as the default instruction-tuning, but remove the instruction portion from each training sample. This ablation model is labeled "No instruction" in Figure 6. This ablation experiment disentangles: (1) training data (present in both Instruction and No instruction), from (2) training LLMs to understand and follow instructions (present only in Instruction).

We use all provided training samples from the Alpaca dataset, thus ensuring that the models are trained on the same amount of data. We observe that brain alignment of the "No Instruction" ablation model increases during fine-tuning but stays lower than its "Instruction" counterpart.

