# OpenReview forum: "Instruction-tuned LLMs with World Knowledge are More Aligned to the Human Brain"
_ICLR.cc/2024/Conference — Submitted to ICLR 2024_

### Official Review · Reviewer_WxVH · 2023-11-01

**Soundness:** 2 fair
**Presentation:** 3 good
**Contribution:** 2 fair
**Rating:** 5
**Confidence:** 4

**Summary:**

This paper presents a study about the relation between LLMs and humans. The motivation is that the instruction-tuned LLMs carry out human instructions better (seems closer to humans). Analysis on both brain activities  shows a closer alignment from LLMs after instruction tuning. The authors also found that the world knowledge and model size are strongly correlated with brain alignment.

**Strengths:**

1. The authors examined two famous and commonly used families of instruction tuned models and find a consistent phenomenon. They also observed the gradual increase in brain score during instruction tuning.

2. The authors studied the models’ fit to both human neural and behavioral data.

**Weaknesses:**

1. It remains unclear in the whole passage that which “internal representations” from LLMs are used, which makes it difficult to reproduce the results.

2. The “world knowledge” part in the BBH dataset is different from the knowledge required in MMLU. The formal consists subsets such as Sports Understanding, Movie Recommendation, and Causal Judgement; While the latter is mainly about disciplinary knowledge such as Anatomy and College Physics. This makes the key term, “world knowledge”, much ambiguous. What knowledge are considered “world knowledge”? Are there any difference between factual, general and disciplinary knowledge?

3. The authors tried to study the effect of world knowledge and model size separately in Section 4.2. However, the two factors are deeply intertwined, given that larger LLMs tend to outperform smaller ones in knowledge-related question answering. The results in Figure 2 also show that model size has even stronger and more significant effect on the brain score. As a result, it cannot be concluded that “world knowledge” is a key contributor to the increase in brain score. Instead, it can be just another indirect effect of the larger model size.

4. The authors use the performance on MMLU and BBH to represent the models’ capability of “world knowledge”. However, performance on these benchmarks is affected not only by the quantity of knowledge that the models possess, but also by their ability to follow instructions. Thus, a higher performance on MMLU and BBH doesn’t necessarily mean that the model has more world knowledge, and the correlation between benchmark scores and brain scores does not necessarily show a link between world knowledge and the fit to human neural data.

5. The authors use the correlation between model per-token perplexity and human reading time to represent the behavioral fit. However, they pointed out in Section 6.2 that this approach is controversial when applied to large Transformer-based models. Thus, the choice of this approach is confusing. Why not use other ways to test the behavioral fit?

6. It is counter-intuitive that factual, domain knowledge can contribute the higher human fit in general reading. In fact, many questions in MMLU are difficult even for most people (e.g., Anatomy, Astronomy, College Physics, ...), and are not going to be retrieved during story reading. It is confusing why the authors choose MMLU as an aspect of the “world knowledge”, and how this can guide Neuroscience research in human language understanding.

7. The three fMRI datasets are in different settings, i.e., reading sentence by sentence, listening the whole passage, and reading word by word, which could bring different activation patterns in the human brain. However, the authors did not discuss the difference between them.

**Questions:**

See the weaknesses part.

---

> ### Author Response · Authors · 2023-11-23
>
> We are happy that the reviewer appreciated that (1) our experiments examine commonly-used model families, and that (2) our results are consistent. Based on their feedback, we ran additional experiments and improved our paper’s clarity. We respond to their points below:
>
> **Q1: The authors tried to study the effect of world knowledge and model size separately in Section 4.2. However, the two factors are deeply intertwined.**
>
> Our work shows that larger LLMs are more aligned to the brain. However, we wish to find out why. What aspects of larger LLMs make them more brain-aligned? Some possibilities: (1) greater world knowledge, (2) better problem-solving abilities, (3) better next-word prediction ability. Our work conducts further experiments to identify the underlying properties of LLMs (beyond model size) that contribute towards greater brain alignment. We show that world knowledge is a key determinant of brain alignment, more than other tested factors.
>
> To show that the plots of world knowledge are not dependent on model size, we wish to highlight that smaller models instruction-tuned to gain greater capabilities (e.g., Vicuna-13B) can achieve greater brain alignment scores than larger models before instruction-tuning (e.g., LLaMA-33B). For reference, here are the average brain alignment values of the models: LLaMA-13B = 0.220, Vicuna-13B = 0.229, LLaMA-33B = 0.227, and Vicuna-33B = 0.232. The Vicuna-13B model has greater brain alignment than LLaMA-33B although it is less than 40% the size. We observe a similar trend when looking at another set of four models: T5-base, Flan-T5-base, T5-large, Flan-T5-large.
>
> To further demonstrate that model size alone does not determine the brain alignment of an LLM, we ran new experiments to evaluate the brain alignment of a randomly-initialized LLaMA-7B. It is a large model with 7B parameters. The randomly-initialized LLaMA was not trained on any data and hence does not contain world knowledge. It also achieves roughly zero accuracy on the BBH and MMLU benchmarks. We observe that the randomly-initialized model achieves lower brain alignment than the trained LLaMA and instruction-tuned Alpaca-7B version. Prior works have also shown that large random embedding models do not achieve high brain alignment scores [1]. These results demonstrate that there are LLM properties aside from model size that contribute significantly to brain alignment.
>
> [1] Schrimpf, Martin, et al. "The neural architecture of language: Integrative modeling …”
>
> **Q2: The authors use the performance on MMLU and BBH to represent the models’ capability of “world knowledge”. However, performance on these benchmarks is affected not only by the quantity of knowledge that the models possess, but also by their ability to follow instructions.**
>
> We agree with the reviewer. Hence, we restricted our selection of LLMs to only two model families. In each model family, the models are trained on a similar format of instruction data, so they have similar ability to follow the instruction format of MMLU and BBH. Thus, any differences in their MMLU or BBH performance would point to differences in the quantity of knowledge they possess, rather than their ability to follow the instruction format of MMLU or BBH.
>
> **Q3: Additional details about our experiment methods**
>
> We thank the reviewer for pointing out areas that require additional clarifications. We have improved our paper’s clarity by adding details on which layers were used to evaluate brain alignment, and the types of world knowledge in BBH and MMLU.

---

### Official Review · Reviewer_KJEe · 2023-11-01

**Soundness:** 2 fair
**Presentation:** 3 good
**Contribution:** 2 fair
**Rating:** 3
**Confidence:** 4

**Summary:**

The paper explores the impact of instruction-tuning on large language models (LLMs) to determine their alignment with the human brain in terms of brain and behavioral alignment. Experimental results from two renowned LLM families indicate that instruction-tuning improves brain alignment by 6.2%, with world knowledge and model size being the primary contributors. However, instruction-tuning does not have a similar effect on behavioral alignment. The authors emphasize the importance of integrating world knowledge in future LLM developments.

**Strengths:**

This paper focuses on the instruction tuning of LLMs, exploring the neuroscience behind language models. This unique perspective advances the understanding of LLMs in the context of human cognition.

The experiment design is intuitive and relatively easy to follow. The experiments are extensive, including 3 datasets for brain and behavioral alignment, respectively.

**Weaknesses:**

Lack of comparative analysis with other tuning techniques such as reinforcement learning from human feedback (RLHF).

The investigation of behavioral alignment is limited. A more comprehensive exploration could offer insights into the discrepancy between brain and behavioral alignments and its implications for LLM development and application.

Other alignment measure methods may be considered to increase the reliability of the results, such as:

Jiaang, Li, et al. "Structural Similarities Between Language Models and Neural Response Measurements." arXiv preprint arXiv:2306.01930 (2023).

Liu, Xu, et al. "Coupling Artificial Neurons in BERT and Biological Neurons in the Human Brain." arXiv preprint arXiv:2303.14871 (2023).

**Questions:**

What is the computational cost of this study? 33B model is quite large, are there any quantization techniques used like LoRA?

May need proofreading: Table 3 and Table 4 in the Appendix have the same caption.

---

> ### Author Response · Authors · 2023-11-23
>
> We are happy that the reviewer thinks (1) our paper presents a unique neuroscientific perspective for understanding LLMs, (2) our experiment design is intuitive and easy to follow, and (3) our experiments are extensive. We respond to their points below and improve our paper based on their feedback:
>
> **Q1: Lack of comparative analysis with other tuning techniques such as reinforcement learning from human feedback (RLHF).**
>
> RLHF is often performed *after* the instruction-tuning phase. Hence, we did not consider it as a comparable alternative to instruction-tuning. In order to perform a comparative analysis between RLHF and instruction-tuning, as suggested by the reviewer, we would need to find LLMs trained with RLHF *instead of* instruction-tuning. Ideally, both LLMs should be trained on the same amount of training data and compute, to allow for a fair comparison. These LLMs also need to be open-source, in order to evaluate their brain alignment. Unfortunately, we could not find such LLMs.
>
> We believe that investigating how RLHF affects the brain alignment of instruction-tuned LLMs constitutes future work. RLHF explicitly aims to align LLMs using human feedback. It would be an interesting direction to see if it also aligns LLMs to human brain activity (brain alignment) and behavior (behavioral alignment).
>
> **Q2: The investigation of behavioral alignment is limited. A more comprehensive exploration could offer insights into the discrepancy between brain and behavioral alignments and its implications for LLM development and application.**
>
> We agree with the reviewer’s point. Hence, we mentioned this in our original submission (and keep it in our rebuttal version too). We included a paragraph in the discussion section 6.2, where we argue for the need to examine more dimensions of behavior. We mention that our work and many prior works compare LM and human next-word surprisal on reading tasks as a measure of behavioral alignment. However, this evaluates only a single dimension of LM and human behavior (per-word perplexity and reading times). We highlight the need to create more benchmarks to expand the dimensions of behavior examined for both LLMs and humans, in order to holistically evaluate LLM behavior, as well as LLM-human behavioral alignment.
>
> Hence, we focus instead on the brain alignment of LLMs.  Our paper focuses on investigating how instruction-tuning affects LLMs from a neuroscientific perspective and the key factors underlying brain-LLM alignment (e.g., world knowledge).
>
> **Q3: 33B model is quite large, are there any quantization techniques used like LoRA?**
>
> For the 33B models, we only performed inference, not training. Hence, LoRA was not used. All our evaluations involve only model inference. This includes the brain alignment evaluations (Pereira2018, Blank2014, Wehbe2014), behavioral alignment evaluations (Futrell2018), and the evaluation of world knowledge (MMLU) and problem-solving abilities (BBH). These evaluations do not involve model training.

---

### Official Review · Reviewer_fAu2 · 2023-11-01

**Soundness:** 3 good
**Presentation:** 3 good
**Contribution:** 3 good
**Rating:** 6
**Confidence:** 3

**Summary:**

This paper investigates the effect of instruction-tuning in the alignment between LLMs’ representations and human language processing. The authors use two types of human data: brain activity patterns (brain alignment) and reading times (behavioral alignment). The brain alignment is defined as the extent to which a linear regression model predicts brain activity patterns using the LLMs’ representations. The behavioral alignment is defined as a correlation between LLM perplexity and human reading time for each word. Through the experiments across 25 vanilla and finetuned models from the T5 and LLaMA families, the authors conclude that instruction tuning improves brain alignment, (2) the performance on the world knowledge-related tasks and model size are correlated with brain alignment, and (3) instruction-tuning and other examined factors are not correlated with behavioral alignment.

**Strengths:**

- Using 25 models and two benchmarking datasets covering various task categories, the authors perform detailed analysis between LLM representations and human brain and behavioral data.
- The discussion includes implications both for NLP and neurosciences along with the literature review, which can encourage interdisciplinary research across both fields.
- The paper is well-written and easy to follow.

**Weaknesses:**

- The authors use models from just two families, T5 and LLaMA. Looking at Figure 2, it seems that LLaMA models do not show a significant correlation between brain alignment and MMLU score, BBH world knowledge, and model size. The results would be more convincing if the authors could use a few more families such as GPT.
- Concerning the tasks related to world knowledge, it appears that these tasks may simply exhibit greater linguistic diversity compared to the other tasks examined. The concept of world knowledge seems somewhat ambiguous, and any clarification could be insightful. For instance, would similar results be observed if more language understanding tasks were added? I was unable to determine how the BBH tasks are categorized into "language understanding" and "world knowledge."

I think that expanding the experiments to address these points could lead to more reliable results.

**Questions:**

- How did the authors determine the category classification for the BBH tasks?
- Is it possible to provide a more detailed analysis regarding world knowledge? Any discussion and additional analysis would be appreciated.

---

> ### Author Response · Authors · 2023-11-23
>
> We are happy the reviewer found that (1) our analyses were detailed, (2) our results had implications for both NLP and neuroscience, and that (3) our paper was well-written and easy to follow. We respond to their points below and improve our paper based on their feedback:
>
> **Q1: The authors use models from just two families, T5 and LLaMA.**
>
> In our work, we evaluate the correlation of brain alignment to other LLM properties, such as world knowledge (MMLU) and problem-solving abilities (BBH). We agree with the reviewer that adding a greater variety of models would strengthen the generalizability of our claims.
>
> However, we realized that the performance on these benchmarks is affected not only by the quantity of knowledge that the models possess, but also by their ability to follow the instruction format of MMLU and BBH. (This point was also mentioned by reviewer WxVH.)
>
> Hence, we restricted our selection of LLMs to only two model families. In each model family, the models are trained on a similar format of instruction data, so they have similar ability to follow the instruction format of MMLU and BBH. Thus, any differences in their MMLU or BBH performance would point to differences in the quantity of knowledge they possess, rather than their ability to follow the instruction format of MMLU or BBH.
>
> If we selected LLMs from many different model families, it would be difficult to control for their ability to follow the instruction format of MMLU and BBH, as they are trained on vastly different instruction formats. This would make it challenging to use their MMLU and BBH results to suggest the quantity of knowledge that the models possess.
>
> However, we overall agree with the reviewer that greater model variety would strengthen generalizability, so we added it to our paper’s section on limitations and future work.
>
> **Q2: How did the authors determine the category classification for the BBH tasks?**
>
> We use the same category classification as the original BBH paper. The BBH paper categorizes tasks into four categories: (1) Algorithmic and Multi-Step Arithmetic Reasoning, (2) Natural Language Understanding, (3) Use of World Knowledge, and (4) Multilingual Knowledge and Reasoning. We add this clarification to our paper.
>
> BBH paper:
> M. Suzgun et al., “Challenging BIG-Bench Tasks and Whether Chain-of-Thought Can Solve Them.”
>
> **Q3: Is it possible to provide a more detailed analysis regarding world knowledge? Any discussion and additional analysis would be appreciated.**
>
> For both MMLU and BBH, we follow their respective papers for how they categorized the world knowledge tasks.
>
> The MMLU paper describes that they design the benchmark to measure knowledge from many domains. It contains 57 tasks, categorized by the subject domain of world knowledge tested: STEM, Humanities, Social Sciences, and Others. The STEM category includes questions on computer science, physics, mathematics, etc. The Humanities category includes questions on philosophy, law, history, etc. The Social Sciences category includes questions on politics, sociology, economics, geography, etc. The Others category includes questions on business topics such as finance, accounting, as well as general knowledge of global facts.
>
> The world knowledge category of BBH contains tasks that test for factual and general knowledge. Tasks requiring factual knowledge include: “Sports Understanding” and “Movie Recommendation”. Tasks requiring general knowledge include: “Causal Judgement”, which tests knowledge about causal-reasoning suppositions, and “Ruin Names”, which requires knowledge about human perception and usage of humor in the English language.

---

### Official Review · Reviewer_co7p · 2023-11-06

**Soundness:** 2 fair
**Presentation:** 1 poor
**Contribution:** 1 poor
**Rating:** 3
**Confidence:** 2

**Summary:**

This paper investigates the correlation between instruction-tuned LLMs and human similarity in the field of neuroscience by examining brain alignment and behavioral alignment. The authors evaluate 25 LLMs on a reading task to identify the effects of instruction tuning LLMs in terms of human language processing. Instruction turned LLMs have higher brain scores than vanilla LLMs, and further analyzed the properties of LLMs that contribute high alignment and found out that the model size and world knowledge are correlated to brain alignments. For the behavioral alignment, there was no correlation between per-word LLM perplexity and per-word human reading times.

**Strengths:**

This paper explains why instruction-tuned LLMs perform better than vanilla LLMs from a neuroscience perspective by measuring brain scores.

**Weaknesses:**

- This paper appears to be a replication of [1,2], specifically focusing on instruction-tuned models. It lacks novelty and originality.
    - Increasing the model size and integrating world knowledge (using a larger training dataset) are not surprising new discoveries for improving language modeling.
    - Additionally, this paper measures the correlation between world knowledge tasks and brain alignment.
    - This paper should demonstrate the effects of contributing factors separately (world knowledge and model size). The plots seem to be dependent on model size.
- The current version of the paper requires further improvement.
    - It lacks details for readers without a background in neuroscience.
        - How is the brain score computed for each model? Does it compute the hidden state of every layer?
    - In Section 4.1, the last paragraph seems to be located too early, making it difficult to understand before explaining the dataset.
    - In Figure 3A, shouldn't the language stimuli be labeled as Futrell2018?
    - Figure 3B appears to be an empty plot.


[1] Schrimpf, Martin, et al. "The neural architecture of language: Integrative modeling converges on predictive processing." *Proceedings of the National Academy of Sciences* 118.45 (2021): e2105646118. \
[2] Oh, Byung-Doh, and William Schuler. "Why does surprisal from larger transformer-based language models provide a poorer fit to human reading times?." *Transactions of the Association for Computational Linguistics* 11 (2023): 336-350.

**Questions:**

- How is the 'No Instruction' model trained in Figure 1D? The Alpaca instruction dataset is formed with both non-empty input fields (instruction, input, output) and empty input fields (instruction, output). Did you only use non-empty input fields and remove the instruction in those cases?
- What aspect do you believe instruction tuning contributes to the correlation between world knowledge and brain alignment?
- In Figure 2, it appears that there are different correlations for each dataset (Pereira2018, Blank2014, and Wehbe2014). Why is Blank2014's correlation so much lower compared to the other two?
- How is word perplexity measured? Could you provide an example of input and the corresponding NWP loss?
    - Since Flan-T5 models are encoder-decoder models, I'm not sure how they are measured differently from decoder-only models. Were the same inputs passed into both the encoder and the decoder?
    - Did the vanilla LLMs also show no correlation?
- Why is there a performance drop when Flan-T5 is fine-tuned on instruction tuning datasets (Alpaca, GPT4ALL, ShareGPT) as seen in Table 5 (Flan-T5-XL results)?

---

> ### Author Response · Authors · 2023-11-23
>
> We are happy that the reviewer appreciated our neuroscientific analysis of instruction-tuning. Based on their feedback, we ran additional experiments and improved our paper’s clarity. We respond to their points below:
>
> **Q1: This paper appears to be a replication of [1,2], specifically focusing on instruction-tuned models. It lacks novelty and originality.**
>
> Our work presents many new insights for researchers studying the parallels between LLMs and human brains: (a) Instruction-tuning aligns LLM representations to human brain activity, and (b) World knowledge is a key factor underlying this alignment. These are both new directions of study that produced new insights. Hence, our work is different from both [1] and [2], which focus on studying how the next-word prediction surprisal of language models relates to human brain activity and behavior.
>
> [1] Schrimpf, Martin, et al. "The neural architecture of language: Integrative modeling …”
>
> [2] Oh, Byung-Doh, and William Schuler. "Why does surprisal from larger transformer-based language models provide a poorer fit to human reading times?."
>
> **Q2: Increasing the model size and integrating world knowledge (using a larger training dataset) are not surprising new discoveries for improving language modeling.**
>
> Our paper is not focused on discovering new methods for improving language modeling. Rather, our paper focuses on evaluating the representational and behavioral similarity of LLMs to the human language system. In particular, we investigate how instruction-tuning affects LLMs from a neuroscientific perspective and the key factors underlying brain-LLM alignment (e.g., world knowledge and model size).
>
> **Q3: The plots of world knowledge (against brain alignment)  seem to be dependent on model size.**
>
> Our work shows that larger LLMs are more aligned to the brain. However, we wish to find out why. What aspects of larger LLMs make them more brain-aligned? Some possibilities: (1) greater world knowledge, (2) better problem-solving abilities, (3) better next-word prediction ability. Our work conducts further experiments to identify the underlying properties of LLMs (beyond model size) that contribute towards greater brain alignment. We show that world knowledge is a key determinant of brain alignment, more than other tested factors.
>
> To show that the plots of world knowledge are not dependent on model size, we wish to highlight that smaller models instruction-tuned to gain greater capabilities (e.g., Vicuna-13B) can achieve greater brain alignment scores than larger models before instruction-tuning (e.g., LLaMA-33B). For reference, here are the average brain alignment values of the models: LLaMA-13B = 0.220, Vicuna-13B = 0.229, LLaMA-33B = 0.227, and Vicuna-33B = 0.232. The Vicuna-13B model has greater brain alignment than LLaMA-33B although it is less than 40% the size. We observe a similar trend when looking at another set of four models: T5-base, Flan-T5-base, T5-large, Flan-T5-large.
>
> To further demonstrate that model size alone does not determine the brain alignment of an LLM, we ran new experiments to evaluate the brain alignment of a randomly-initialized LLaMA-7B. It is a large model with 7B parameters. The randomly-initialized LLaMA was not trained on any data and hence does not contain world knowledge. It also achieves roughly zero accuracy on the BBH and MMLU benchmarks. We observe that the randomly-initialized model achieves lower brain alignment than the trained LLaMA and instruction-tuned Alpaca-7B version. Prior works have also shown that large random embedding models do not achieve high brain alignment scores [1]. These results demonstrate that there are LLM properties aside from model size that contribute significantly to brain alignment.
>
> [1] Schrimpf, Martin, et al. "The neural architecture of language: Integrative modeling …”
>
> **Q4: How is the 'No Instruction' model trained in Figure 1D? Did we use all the training data from the Alpaca dataset?**
>
> We used all provided training samples from the Alpaca dataset. This is to ensure that the ablation model is trained on the same data as the instruction-tuned Alpaca-7B, with the only difference that the ablation model does not receive the “instruction” portion during training. We added this clarification (that we use all the provided training samples) to our paper.
>
> **Q5: Additional details about our experiment methods**
>
> We thank the reviewer for pointing out areas that require additional clarifications. We have improved our paper’s clarity by adding details on the methods used for computing brain alignment, measuring next-word prediction loss, and running our ablation experiments.

---

### Meta-Review · Area_Chair_YcG5 · 2023-12-12

**Metareview:**

In this paper, the authors study the effect of instruction-tuning on the alignment between LLMs’ representations and human language processing. Two types of human data, including brain activity patterns (brain alignment) and reading times (behavioral alignment), are investigated. Experiments across 25 vanilla and finetuned models from the T5 and LLaMA families show that instruction turned LLMs achieve higher brain scores than vanilla LLMs, world knowledge and model size are correlated with brain alignment, and instruction-tuning is not correlated with behavioral alignment. The analysis from neuroscience perspective is interesting and valuable.

Models from just two families, T5 and LLaMA, are used, and the results on LLaMA do not show significant correlation.  The authors should use a few more models from the GPT family to make the results more convincing. In addition, the presentation of the paper should be significantly improved, including a clearer description of the contribution/novelty and a clearer intro of related neuroscience background.

**Justification For Why Not Higher Score:**

Models from just two families, T5 and LLaMA, are used, and the results on LLaMA do not show significant correlation.  The authors should use a few more models from the GPT family to make the results more convincing. In addition, the presentation of the paper should be significantly improved, including a clearer description of the contribution/novelty and a clearer intro of related neuroscience background.

**Justification For Why Not Lower Score:**

N/A

---

### Decision · Program_Chairs · 2024-01-16

Reject